# Impingement of systems containing suspending droplets and coming ones at the nanoscale

**Liwei Sun[1], Yi Guo[2], Tianshi Yang[3], Xudong Ma[1]\***

**1** School of Mechanical Engineering, Changchun Technical University of Automobile, Changchun, China, **2** Jilin Polytechnic of Water Resources and Electric Engineering, Changchun, China, **3** Jilin Communications Polytechnic, Changchun, China

\* n732589404@163.com

## Abstract

Impingement of binary droplets at the nanoscale is very important and is ubiquitous in daily life and practical applications, such as lab-on-a-chip and digital microfluidics. The current work performs molecular dynamics (MD) simulations to simulate free evolution of impacting deposited droplets with incoming ones with different height differences. The representative dynamics under different given conditions have been directly captured with the help of MD simulations. We observe that the targeted systems initially experienc primary spreading followed by secondary spreading, and ultimately, merged droplets can deposit upon the surface or form bouncing ones. Through recordation of the maximal spreading factor over primary and secondary spreading, the law of spreading at various separated distance ($D_{sep}$) and impacting Weber number ($We$) is investigated. The increasing $D_{sep}$ can obviously postpone the collision obviously while the increasing $We$ allows the collision to be advanced. The external factor of $D_{sep}$ can not affect the primary spreading but shows a significant influence on the secondary spreading. The prefactor of the secondary spreading at low-velocity impingement is 0.06 whose value is the same as that for primary spreading. On further increasing $We$, the high-velocity impingement alters the secondary spreading with prefactor increasing from 0.06 to 0.09. This is attributable to the fact that the collision between incomplete retracting droplets and incoming ones resists the secondary spreading. But at high velocity, the pattern of incomplete-retraction collision can promote the secondary spreading. The variation of contact time at a wide range of $D_{sep}$ and $We$ is recorded, and the effect of these two mentioned parameters is investigated. This work can help researchers to understand the impact of binary nanodroplets with various $D_{sep}$, which is very suitable for guiding the practical applications that need different impacting conditions.

**Data availability statement:** All relevant data are within the manuscript and its Supporting Information files.

**Funding:** The author(s) received no specific funding for this work.

**Competing interests:** The authors have declared that no competing interests exist.

## 1. Introduction

Recently, the topic of impingement of water droplets upon solid surfaces has received a great deal of attention, because it is not only the most common in nature and daily life but also has potential applications in industry, agriculture, and so forth [1–8]. Since the first work had been done to observe impacting droplets, many researchers followed this step to devote themselves to exploring impingement of water droplets through theoretical analyses, experimental tests, and numerical simulations [9–12]. Generally, the impacting droplets may exhibit spreading, recoiling, bouncing, and splashing behaviors at room temperature [13–15]. Several factors take control of impacting dynamics, including physical properties of droplets (viscosity, surface tension, and size), surrounding environment (pressure, humidity, and temperature), and physical and chemical properties of targeted surfaces (wettability, surface morphology, etc).

To account for numerous influencing factors, a series of dimensionless parameters have been used to help researchers investigate impacting dynamics. Two representative dimensionless numbers, Weber number ($We = \rho D_0 V_0^2/\gamma$) and Reynolds number ($Re = \rho D_0 V_0/\mu$), are generally introduced to describe the different dominant force. The $We$ represents the ratio of inertial force to capillary force, and the $Re$ is used to express the ratio of inertial force to viscous force. Where $\rho$, $\gamma$, and $\mu$ are density, surface tension, and viscosity, respectively, $V_0$ is the given impacting velocity, and $D_0$ is the diameter of impacting droplets without any deformation. Therefore, the dynamic evolution of impacting water droplets can be well described by involving these dimensionless numbers. For example, the increase in $We$ can enlarge the deformation of impacting droplets and increase the opportunity to induce splash while the increasing $Re$ prevent these behaviors from occurring. The energy analysis is always adopted to account for different behavior from spreading and retraction to bouncing and splash [16,17]. The process of spreading is driven by energy conversion from kinetic energy to surface energy, and the increasing surface energy can be stored in droplets' deformation. In contrast, the surface energy can be released to make droplets retract, and finally, droplets retract to deposited or bouncing ones. Apart from dimensionless parameters, there are several other important parameters for characterizing the process of droplets' impingement, including maximum spreading diameter, bouncing height, contact time, restitution coefficient, and so forth [18–20]. The contact time and bouncing height are indispensable in the analysis of dynamic behaviors of impacting droplets on superhydrophobic surfaces, which can directly reflect bouncing kinetic characteristics. The shorter contact time and higher bouncing height are very desirable for anti-icing characteristics; otherwise, impacting droplets would adhere to solid surfaces. When a droplet with $D_0$ impacts upon a surface, it spreads rapidly and attains a pancake shape with $D_{max}$, at which time kinetic energy can be completely transformed into surface energy. To minimize the size effect, the maximum spreading diameter is generally involved in a dimensionless formation and is expressed as $\beta_{max} = D_{max}/D_0$. The $\beta_{max}$ is considered a decisive factor for precise operation ininkjet printing or the absorption rate of pesticide spraying. Commonly, the value of $\beta_{max}$ over the course of droplets' impingement can be predicted using a

scaling law [21]. According to energy conservation, the maximal spreading factor can be deduced as $\beta_{max} \sim We^{1/4}$ [22]. For high-viscosity fluids, the $\beta_{max}$ should be further modified by integrating $Re$ into the scaling law because the viscous boundary must be considered [23–25]. Ref. 25 proposed a scaling law of $\beta_{max} \sim Re^{1/5}$ to predict the maximum spreading factor, which is found to agree well with experimental data over a full spectrum of Weber number. The coefficient of restitution, $\varepsilon_{co}$, can reflect bouncing kinetic characteristics in another way, which is defined as the ratio between bouncing energy to impacting energy. Ref. [26] reported that, for low-viscosity fluid, the $\varepsilon$ can be predicted using a power law with respect to the Weber number. The $\varepsilon_{co}$ strictly obeys a law of $We^{-1/4}$ at low $We$ range, while such a law is replaced by $\varepsilon_{co} \sim We^{-2/3}$ after increasing the Weber number to a relativey high value.

Nowadays, the nano-scale impingement gradually becomes a hot topic owing to its special application in nanofields, such as nanospray, nano ink-jet printing, lab-on-a-chip, and so forth [27,28]. Nevertheless, it is difficult to investigate nanoscale impingement by conventional methods because they are limited by both time scale and space scale. With the help of molecular dynamics (MD) simulations, the dynamic evolution of impacting nanodroplets with tens of nanometers can be observed directly. Therefore, MD simulations provide researchers an opportunity to obtain natural insights into nanoscale dynamics [29–32]. The nanoscale impingement shows many different dynamic behaviors together with new mechanisms. For example, the nanoscale water droplet is regarded as a natural "high-viscosity fluid", which has proved by calculating relevant Ohnesorge number. The Ohnesorge number exhibits a significant increase from a magnitude of $10^{-3}$ to 1, indicating that the viscous force can no longer be ignored [33]. Using MD simulations, Ref. [34] reported that although the size of impacting droplets reduces to nanoscale, the traditional outcomes, including deposit, bouncing, and breakup, can still be observed. The impingement also introduces some novel outcomes, like hole rebound, partial-reboundsplash, and rebound splash, which can never be seen at the millimeter scale. For macrosystems, the energy dissipation only occurs with a very thin boundary, so that the dissipation is out of consideration. However, for nanoscale fluids, the velocity can almost permeate the whole nanodroplets, which causes a great increase in energy dissipation, as described in Ref. [35]. Up till now, the impingement of nanoscale droplets under different conditions is relatively well understood. However, the impingement of multiple droplets is more close to practical applications, which is more complex and requires more effort to understand. For collision of macrodroplets, the dynamic behavior has been captured, which initiates with approach process followed by permanent coalescence, bounce, and breakup [36,37]. Previous studies demonstrated that bounce only occurs in a special situation where the input energy exceeds the surface energy from liquid bridge [36]. The bouncing behavior disappears for nano-scale collision because of significant increase in viscosity and surface-to-volume ratio [36]. Intriguingly, the bounce after separated droplets coalesce can also be observed for nanodroplets with about 100 nm. Thus, the viscous force gradually becomes one of the most important parameters if the size of impacting droplets reduces to below 100 nm [38]. Moreover, the collision process of two identical droplets at the nano-scale in a suspended state has been investigated [39]. Up till now, how the solid surface affects the dynamic evolution of binary nanodroplets is incomprehensible. To fill this gap, we perform MD simulations to simulate the impingement of binary nanodroplets with different height differences on superhydrophobic surfaces. The free evolution of impacting binary droplets can be directly captured through extracting snapshots from MD simulations. The important parameters of targeted systems, including maximum spreading factor during primary and secondary spreadings and contact time, are investigated and discussed. The present work can give insights into the impingement of binary droplets at various separated distance and $We$, which guides a situation where a practical application needs to apply impacting systems containing binary nanodroplets.

## 2. Simulation method

In this work, MD simulations (LAMMPS) are used to construct an initial configuration and simulate the free evolution processes at different given conditions, including different height differences and Weber numbers. In the first step, we establish the initial configuration, as shown in **Fig 1**. Herein, two droplets are static and suspended in a vacuum environment,

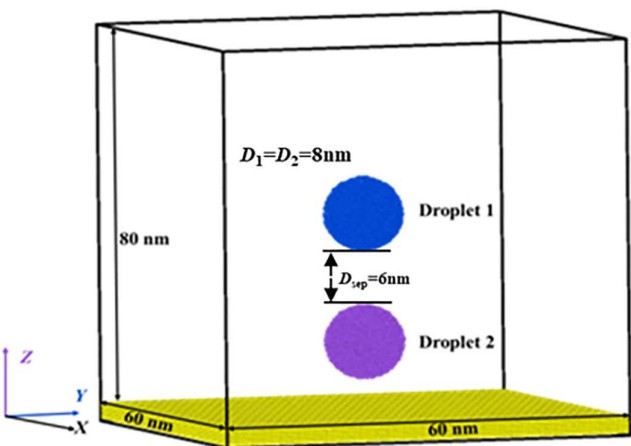

**Fig 1. The targeted system of the present work contains two identical nanodroplets with a diameter of 8 nm.** Two droplets are put into a vacuum environment with a separated distance of 6 nm.

and the solid surface consists of Pt atoms. Droplets are completely identical with a diameter of 8 nm, containing 8900 water molecules, and the center-of-mass coordinates of two suspended droplets are (0 nm, 0 nm, 12 nm) and (0 nm, 0 nm, 26 nm), respectively. The separated distance between droplets is $D_{sep} = 6$ nm. To prevent the undesirable deformation of Pt surfaces, we adopt an artificial virtual spring imposed on each metal atom, and thus, the Pt atoms can be fixed at their initial position, which is widely adopted in previous investigations [40,41]. The simulated system uses periodic boundary conditions in each direction with dimensions of 40 nm, 40 nm, and 60 nm in x-, y-, and z- directions.

We perform a mW (monoatomic water) model to describe the intermolecular interactions between water molecules. Such a model is coarse-grained, which uses a large group for simplifying the structure of water molecules. We chose this from numerous models (such as TIP4P and SPC/E models) to reduce the computing cost because of its monatomic formation [42,43]. The physical property of water, such as density and surface tension, still has the ability to be reproduced [42]. Owing to omission of the hydrogen atom reorientation, the viscosity of the model is $\mu = 283.7 \mu Pa.s$, which is three times lower than the experimental value [42]. The van der Waals force for both water-Pt and Pt-Pt can be described by the Lennard-Jones 12−6 potential, expressed as:

$$U_{\mathrm{LJ}}(r) = 4\varepsilon \left[ \left(\frac{\sigma}{r}\right)^{12} - \left(\frac{\sigma}{r}\right)^{6} \right], r < r_{\mathrm{cut}}$$

(1)

where $r$ is a distance separating two adjacent atoms, $\varepsilon$ is the depth of the potential wall, $\sigma$ is the zero-crossing distance, and $r_{cut}$ is the cutoff distance with a value of 1 nm [43]. The interaction between water molecules is described by parameters of $\varepsilon_w = 0.26838$ eV and $\sigma_w = 0.23925$ nm, whose values change to $\varepsilon_P = 0.69375$ eV and $\sigma_P = 0.247$ nm to describe Pt-Pt interaction. Previous studies demonstrated that these parameters are effective in describing the dynamic behavior of nanoscale droplets [44,45].

After that, we relax the system for 1 ns with a time step of 1 fs to obtain the equilibrium state. To achieve this purpose, the system is run in the NVT ensemble at 300 K over a pre-equilibrium process using Nose–Hoover thermostat [46–48]. Subsequently, two droplets are run in the NVE ensemble for another 1 ns in productive processes. Impacting droplets are endowed with a series of vertical velocities to impact suspending ones in an attempt to observe the dynamic evolution from coalescence, spreading, retraction to bounce, and breakup. The wettability of solid surfaces is generally controlled by a parameter of $\varepsilon_{w\text{-}Pt}$, which can express the interaction between water and surfaces. Hence, we choose two values of

$\varepsilon_{\text{w-Pt}}$ as 0.0108 eV to 0.0045 eV to construct different intrinsic wettability from hydrophobic to superhydrophobic. To do so, we can investigate how $D_{\text{sep}}$ and *We* affect the dynamic evolution of targeted systems. During the productive process, the system is controlled by NVE resembling, and the coming droplet is endowed with different impacting velocities. The velocity-Verlet algorithm is applied to update the position and velocity of each particle with a time step of 0.002 ps.

For impingement of a water droplet with 20 μm upon on superhydrophobic surfaces, Ref. [49] shows that the droplet starts to spread after it interacts with the solid surface, subsequent retraction occurs with a liquid nail pinning on the surface, and ultimately water droplet leaves off from the surface, as shown in **Fig 2a**. Our simulated results of the impacting water droplet with a diameter of 8 nm is illustrated in **Fig 2b**. These snapshots are consistent with what we have observed in experimental tests (see **Fig. 2a**). The tiny deviation may come from the following two reasons. First, the present system adopts a nanoscale droplet with only 8 nm while the size of the impacting droplet in experiment is tens of micrometers. Another one is that the environment of experimental observation is normal pressure and temperature, while MD simulations are performed with a vacuum environment.

## 3. Results and discussion

To investigate the dynamics of impacting binary nanodroplets under different given conditions, we start from observation of the free evolution of the targeted system with a separated distance of $D_{\text{sep}} = 6$ nm at $We = 1.21$. We take the corresponding dynamic evolution as representative evolution progress. As shown in **Fig 3a**, owing to the existence of a height difference, two droplets undergo completely different dynamic behaviors. The droplet below first comes into contact with the surface at $t = 14$ ps. The kinetic energy makes the below droplet spread and reach its maximum deformation at 28 ps. After that, the droplet retracts towards the center due to the release of the stored surface energy. The behavior is the same as the normal impingement and is named as the primary spreading of targeted system. The variation of mass center of these two droplets is then recorded to help us further understand the dynamic evolution of impacting binary droplets, as shown in **Fig 3b**. For the impacting droplet below, the mass center initiates with a reduction of its value to a minimum value over spreading stage and increases reversely when the droplet starts to retract ($t < 50$ ps, see **Fig 3b**). A different scenario occurs at the moment when two droplets contact with each other at $t = 52$ ps in **Fig 3a**. Intriguingly, the collision of separated droplets involves further increasing the value of the mass center of droplet 2 rather than a reduction of its value. The

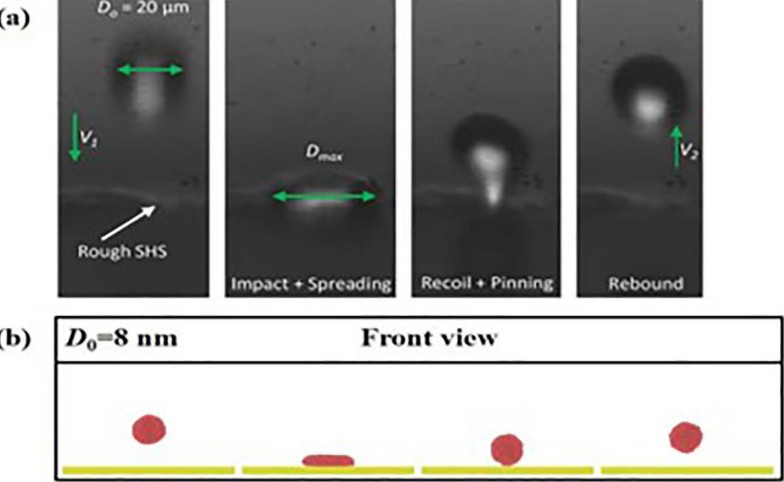

**Fig 2. (a) Experimental results of impacting droplet with 20 μm upon a superhydrophobic solid surface. (b)** Sequential images of impingement of a nanodroplet with diameter of 8 nm from MD simulations.

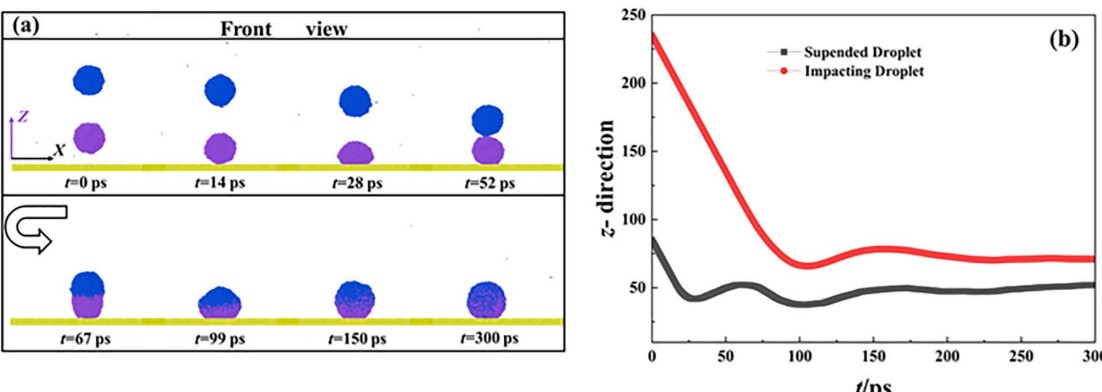

Fig 3. (a) Impingement of binary nanodroplets with $D_{sep} = 6$ nm at $We = 1.21$ (b) The corresponding variation of mass center of each nanodroplet over impingement.

coalescing process makes the liquid bridge gradually expand due to the pressure difference between the inner and outer liquid bridge [50]. The growth of the liquid bridge and the incomplete retraction of the below droplet can release a small amount of surface energy, which is responsible for increasing mass center (see 52–67 ps, **Figs. 3a** and **3b**). The turning point of the droplet in purple starts under the action of kinetic energy provided by the incoming one. The merged droplet spreads on the surface and reaches another maximum deformation, namely secondary spreading of the targeted system. The retraction occurs once again, and the merged droplet ultimately forms a stable deposited state with a constant value of mass center after several vibrations.

In this work, the increment of speed of impacting droplets is 100 m/s, and the corresponding Weber bumber can be calculated based on $We = \rho D_0 V_0^2 / \gamma$. When $We$ increases to 30.18, the dynamic evolution of impacting systems from primary to secondary spreading is very different from low-velocity impingement. The snapshots of the corresponding impingement process are illustrated in **Fig 4a**. The below droplet fast involves a pancake shape at its primary spreading (see 11 ps, **Fig 4a**). Importantly, the collision of two droplets is greatly advanced soon after the below droplet exhibits the maximal deformation at about 15 ps. The mass centers of the original two droplets quickly coincide with each other during the

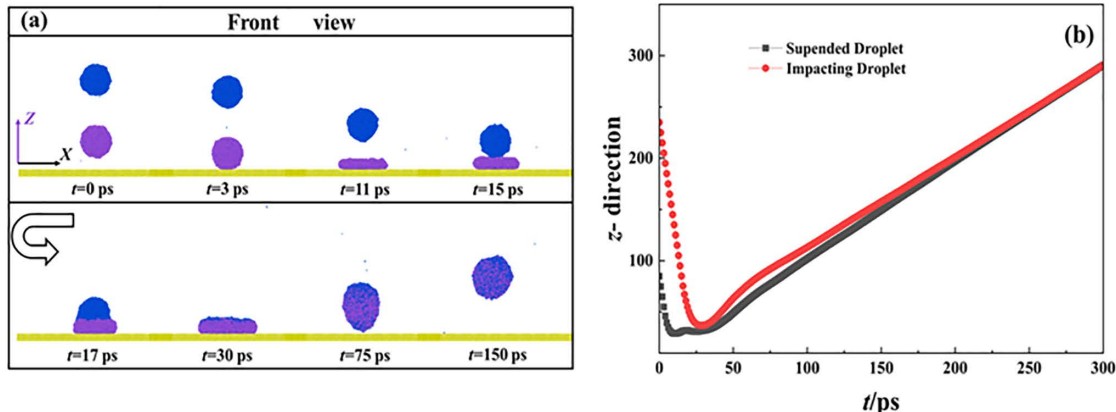

Fig 4. (a) Free evolution of impingement of binary nanodroplets at $We = 30.18$ and (b) corresponding variation of mass center of two droplets as a function of time.

secondary spreading (30 ps, **Fig 4b**), and a high degree of interdiffusion can be observed. The increasing Weber number provides more surface energy stored in droplet's deformation. Therefore, the merged droplet can overcome the work of adhesion and bounce off the solid surface after 75 ps. The mass center of the merged droplet increases at an invariant rate during the bouncing stage because gravity can be ignored, leading to a constant velocity of the merged droplet rising.

At high *We* range of *We* > 100, the unstable dynamics occur and the corresponding snapshots are shown in **Fig 5**. The primary spreading greatly reduces the thickness of the water film at *We* = 120.8. The dramatic progress even forms a very thin spreading droplet with several nanometers in an irregular contour (see 8 ps, **Fig 5a**). The secondary spreading allows instability to further develop, which induces a huge hole within the droplet at 32 ps (**Fig 5a**). The impacting droplet can bounce again with a closed-up hole. At an extremely high value of *We* = 146.08, the breakup occurs and the targeted system eventually produces many small fragments dispersed in the simulated box, as shown in **Fig 5b**. For these two impacting cases, many gas molecules escape from impacting droplets because the high-energy molecules can overcome the constraint of the bulk droplets.

The topic of the current work is to investigate how varying height differences affect the free evolution of targeted systems. To achieve this purpose, we record the instantaneous variation of dimensionless spreading diameter, $\beta$, as a function of time at various $D_{\text{sep}}$ and *We*, as shown in **Fig 6**. For the special binary droplets' systems, the $\beta$ is expressed as $\beta = D(t)/D_0$. The curve of $\beta$ has two peaks, which corresponds to the maximum spreading state at processes of

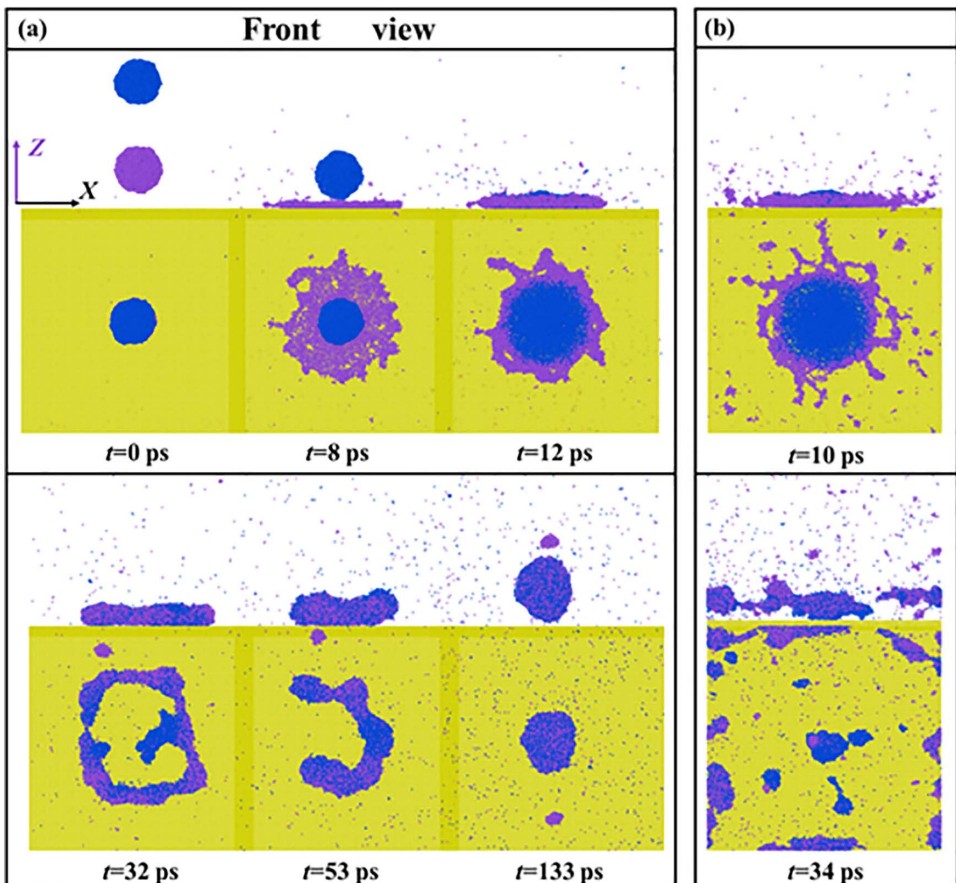

**Fig 5. Free evolution of unstable dynamics of the targeted system at (a)** *We* = 120.8 **and (b)** *We* = 146.08.

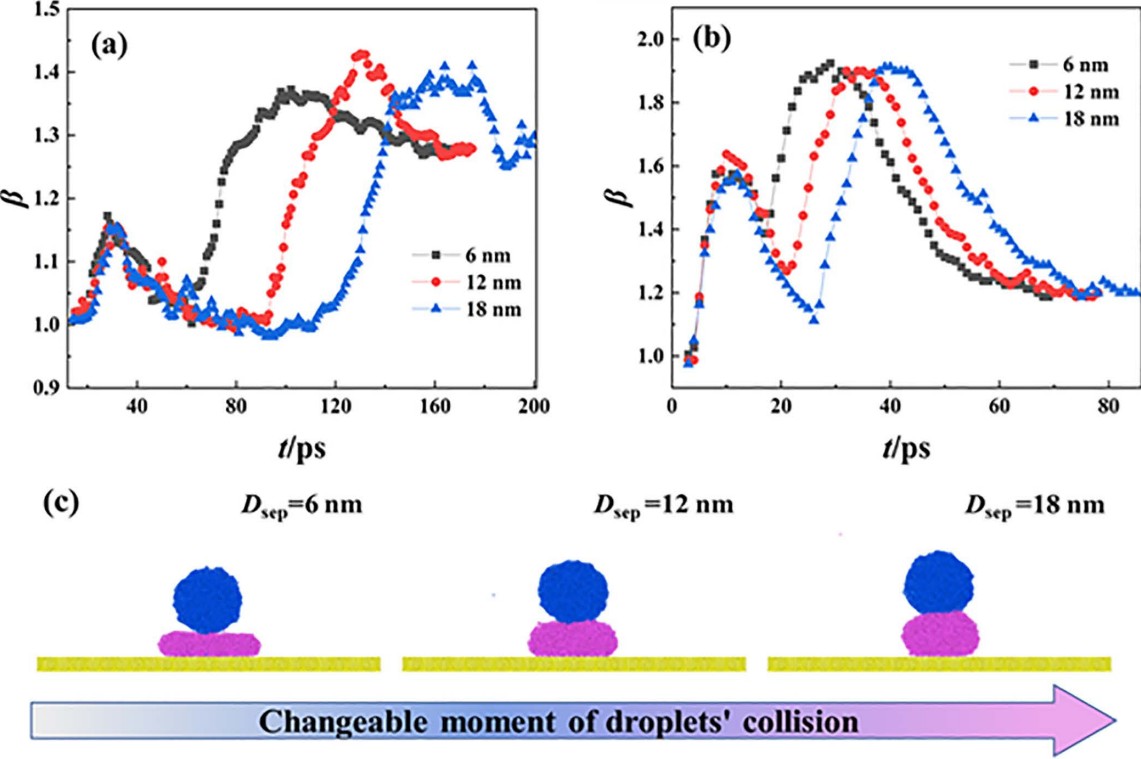

**Fig 6. Variation of dimensionless spreading factors of targeted systems as a function of time at (a) We = 1.21 and (b) We = 30.18. (c)** The evolution of droplets' collision with $D_{sep}$ = 6 nm, 12 nm, and 18 nm at We = 30.18.

primary spreading and secondary spreading. When the Weber number is selected, the primary spreading has a consistent trend of change with reaching the same maximum primary spreading factor, $\beta_{pri, max}$, when $D_{sep}$ changes in range of 6 nm < $D_{sep}$ < 18 nm. Therefore, the primary spreading should be independent of the impingement of incoming droplets. Generally speaking, the increasing $D_{sep}$ postpones the turn point of these curves because the time required for nanodroplets' contact becomes longer. The turning point of the curve indicates that the secondary spreading occurs, and the increasing rate of each curve at the same We is almost similar. Therefore, the inertia force plays a predominant role in secondary spreading rather than the moment of droplets' collision. As shown in **Fig 6a**, for low-velocity impingement, the value of $\beta_{sec, max}$ at $D_{sep}$ = 6 nm is lower than the other two cases. We carefully observe the corresponding snapshots in **Fig 3a** to look for the underlying reason. The retraction of below droplet upon collision is a dilemma of the conflict of the secondary spreading. As such, the secondary spreading must consume a degree of extra energy, and thus, the relevant value of $\beta_{sec, max}$ at $D_{sep}$ = 6nm is relatively low. When $D_{sep}$ increases to 12 nm and 18 nm, the collision occurs after the below droplet achieves its retracting behavior completely. Hence, the primary spreading no longer affects the progress of the secondary spreading any more, resulting in the same $\beta_{sec, max}$. The increasing We to 30.18 leads $\beta_{pri, max}$ and $\beta_{sec, max}$ to increase significantly due to high input energy, as shown in **Fig 6b**. The moment of collision occurring continuously is postponed by increasing $D_{sep}$ at We = 30.18, as schematically illustrated in **Fig 6c**. However, the effect of changeable $D_{sep}$ on the $\beta_{sec, max}$ is also weakened. The increasing We makes the inertia force dominant and the energy from the retracting stage is inappreciable, unlike low-velocity impingement.

To overall understand incomplete retraction, we define a dimensionless parameter as $h^* = h_{sp}/h_{sta}$ to help us quantificationally express incomplete degree of merged droplets' retraction. Here, the $h_{sp}$ is the height of mass center of below

droplets at the monement when two separated droplets just contact with each other and $h_{sta}$ is the height of natural spreading of water droplets with a diameter of 8 nm. Therefore, the value of $h^*$ has two limited values: $h^* = 1$ indicates that the collision starts with below droplets being in a complete retraction state, while $h^* = 0$ stands for collision at complete spreading of below droplets with insignificant thickness. As shown in **Fig 7a**, the increasing value of *We* enlarges the degree of incomplete retraction, as mentioned above. On the other hand, the increasing $D_{sep}$ has the opposite tendency to reduce the deformation degree of spreading droplets once the impacting Weber number is specific, especially for $D_{sep} = 18$ nm. This is due to the fact that the increasing $D_{sep}$ makes the spreading droplet have more time to recover its deformation. However, the effect of $D_{sep}$ and *We* on varying $h^*$ gradually weakens at higher *We* value, and thus, the incomplete degree initially increases and approaches to a limition whose value is about 0.2. The spreading water film only contains several molecular laryers in such situations. According to observation from **Fig 6**, we find that the state of incomplete retraction may affect merged droplets' spreading. Therefore, we map a phase diagram to inveistage whether the incomplete retraction has the influence on the secondary spreading, as shown in **Fig 7b**. The incomplete retraction can suppress the secondary spreading when *We* and $D_{sep}$ are relatively low, see the regime of suppressive secondary spreading. On futher increasing *We*, the secondary spreading is irrelevant to *We* and $D_{sep}$ because of the impacting process being dominated by inertial force, see regime of suppressive secondary spreading.

Next, we study the variation of maximum spreading factor that is directly related to precision of ink-jet printing. For the spreading of macrodroplets, the $\beta_{max}$ is only related to the dimensionless parameter of *We*. Owing to the non-negligible viscous effect, the parameter of *Re* must be considered over the course of spreading, as described in Refs. 9 and 36. According to the energy conversion, the relationship between kinetic energy and surface energy during droplets' impingement follows $\gamma D_0^2 \sim E_K V_0^2$. Where $D$ is the diameter of the merged droplet and $V_0$ is the given velocity of impacting droplet. Therefore, the maximum spreading factor can be derived as $\beta_{max} \sim We^{1/2}$. Moreover, the viscous dissipation should be scaled as $Re^{1/5}$, and thus, the $\beta_{max}$ at the nanoscacle should be $\beta_{max} \sim We^{1/2} Re^{1/5}$ by involving a comprehensive influence for both *We* and *Re*. To obtain deep insights into the variation of $\beta_{max}$, we record values of $\beta_{pri, max}$ and $\beta_{sec, max}$ at different given conditions, as shown in **Fig 8**. Herein, the $D_{sep}$ is in a range between $D_{sep} = 6$ nm and 18 nm and *We* ranges from 1.21 to 99.23. As shown in **Fig 8**, the value of $\beta_{max}$ is almost equal for a specific Weber number regardless of how $D_{sep}$ varies. Most of data extracted from MD simulations agree well with the law of $\beta_{max} \sim We^{1/2} Re^{1/5}$, besides some extreme cases of the secondary spreading. In addition, the same value of $\beta_{pri, max}$ indicates that the collision occurs after the below droplets have achieved

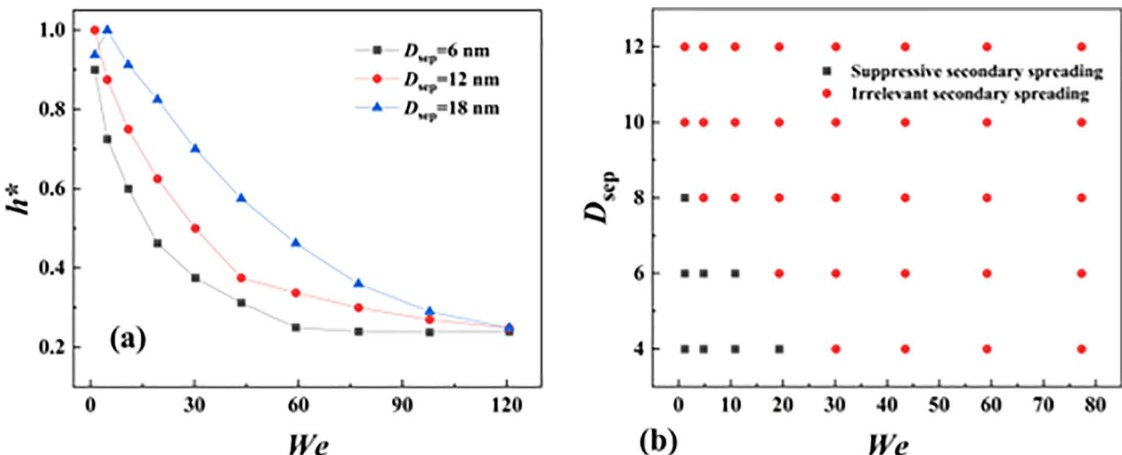

**Fig 7. (a) Variation of $h^*$ as a function of *We* at different values of $D_{sep}$. (b)** Phase diagram containing irrelevant and suppressive secondary spreading regimes with respect to $D_{sep}$ and *We*.

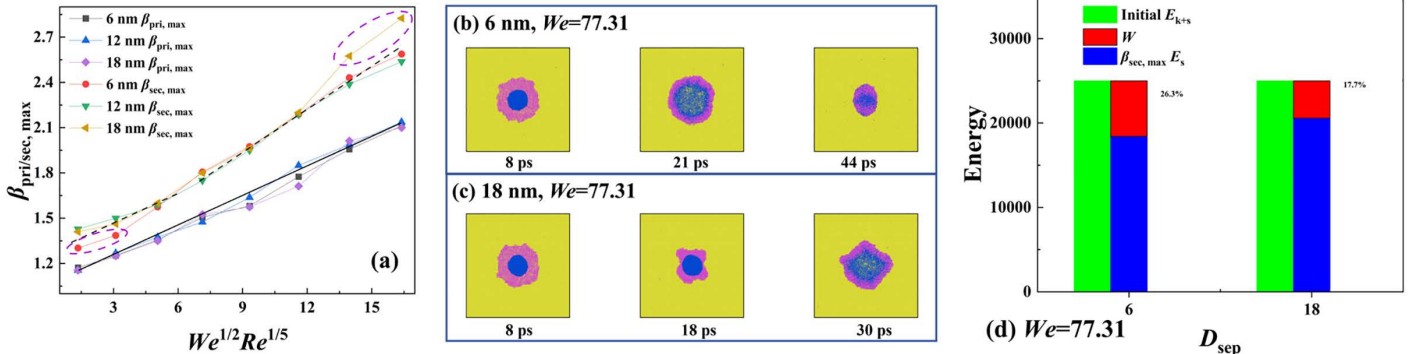

**Fig 8. (a) Variation of $\beta_{pri, max}$ and $\beta_{sec, max}$ at various $D_{sep}$ as a function of $We^{1/2}Re^{1/5}$.** The free evolution of impacting binary nanodroplets at $We = 77.31$ under **(b)** $D_{sep} = 6$ nm and **(c)** $D_{sep} = 18$ nm. **(d)** The proportion of energy dissipation at $We = 77.31$ and $D_{sep} = 6$ nm and 18 nm.

spreading dynamics, even when the $D_{sep}$ reduces to 6 nm and the velocity is increased to 800m/s. The prefactor of the law of the primary spreading is calculated as 0.06, see solid line in **Fig 8a**. The situation becomes more complex when the merged droplet experiences secondary spreading, and variation of the $\beta_{sec, max}$ can be further divided into two parts. In the first part from $We = 1.21$ to 10.87, the increasing rate of $\beta_{sec, max}$ is consistent with the primary spreading, apart from impingement at $D_{sep} = 6$ nm. The low value of $\beta_{pri, max}$ at $D_{sep} = 6$ nm is attributable to the collision occurring between a coming droplet and an incomplete retracting one. In part two, the targeted systems have a profound influence on the secondary spreading when $We$ increases above 11. The impingement of coming droplets can promote the spreading of targeted systems, with the prefactor increasing from 0.06 to 0.09 (see dotted line). The collision also occurs in a formation of incomplete retraction, but the recovering velocity upward can not resist the falling droplet. The fast deformation over secondary spreading can significantly increase the distance among water molecules, leading to a weak van Der Waals force. Hence, the incomplete-retraction collision, on the other hand, changes to a positive role to promote spread. The high-velocity impingement at high $D_{sep} = 18$ nm can significantly increase value of $\beta_{sec, max}$. To account for this special behavior, we extract snapshots of impacting binary nanodroplets with $D_{sep} = 6$ nm and 18 nm at $We = 77.31$, as shown in **Fig 7b** and **8c**. At $D_{sep} = 6$ nm, the collision starts from the below droplet just entering the retracting stage, like normal secondary spreading, as shown in **Fig 8b**. We next calculate the enrgy variation from intial retraction of below droplet to collision between two separated nanodroplets, as shown in **Fig 8d**. Due to the non-negligible viscous effect, the retraction also induces a certain degree of energy dissipation, whose value accounts for 16.3 percent of the total energy. The collision at $D_{sep} = 18$ nm occurs at the final stage of below droplet's retraction. The droplet in such a situation involves a very strangeshape (**Fig 8c**), i.e., the asymmetrical retraction. The strange shape is found to significantly reduce the energy dissipation over the couse of below droplet's retraction (see **Fig 7d**). This may be due to the fact that the fast deformation increases the rigidity of the spreading droplet together with energy dissipation. On the other hand, the postponed coalescence of merged droplets can drops the mentioned rigidity and aggravates the asymmetrical evolution, leading to deviation from the scaling law.

With a progressive increase in separating distance $D_{sep}$, the moment of droplets' collision exhibits hysteresis continuously. When $D_{sep}$ increases to 70 nm, the collision even occurs after complete bounce of the below droplet, and the corresponding snapshots are illustrated in **Fig 9**. The droplet below comes into contact with the surface, followed by spreading, retracting, and bouncing. During this period, the upper droplet falls and eventually coalescence starts at $t = 55$ ps. Owing to the existence of energy dissipation, the coefficient of restitution is less than unity. Hence, the kinetic energy of the bouncing droplet is lower than the incomingones, and subsequently undergoes the secondary spreading, namely collision with bouncing droplet. This special dynamics is not our main purpose of the present work, which may be discussed in detail in our future work.

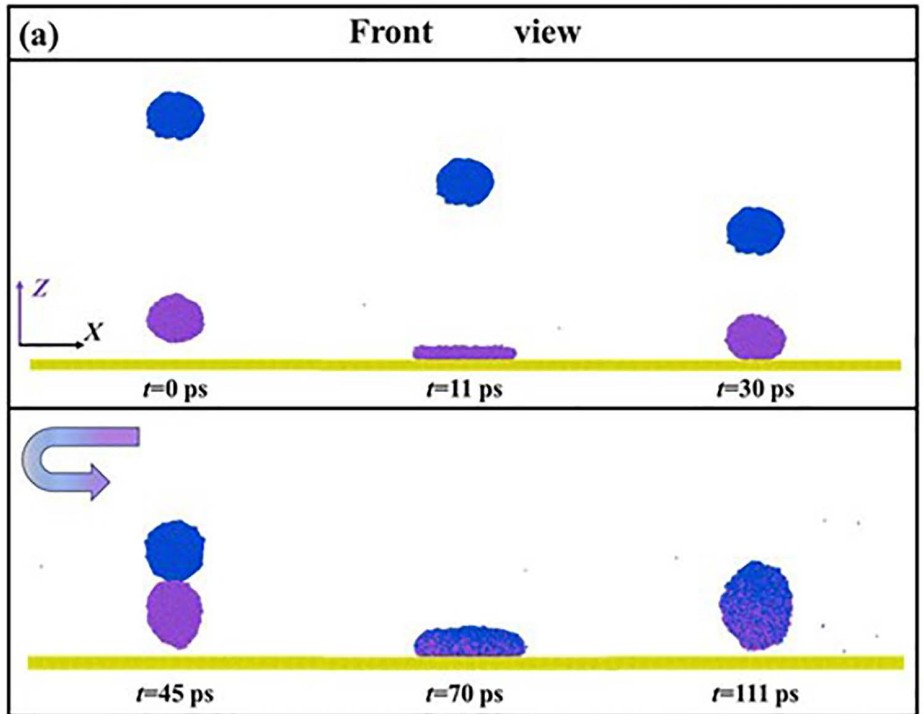

**Fig 9. The collision of separated nanodroplets after the below droplet completely bounces off from solid surface at *We*=42.65.**

Ultimately, we focus on the variation of contact time, $t_c$, on superhydrophobic surfaces at a wide range of Weber numbers. The diagram of $t_c$ with respect to $D_{sep}$ and *We* is recorded and illustrated in **Fig 10a**. For targeted systems, the $t_c$ is defined as a time span from the below the droplet first comes into contact with the solid surface to the bouncing of the merged droplet. We observe that the $t_c$ initially reduces, gradually plateaus to a constant value, and increases with further increasing *We*. The reduction of the $t_c$ is well recognized, resulting from the increasing stored surface energy in droplets'

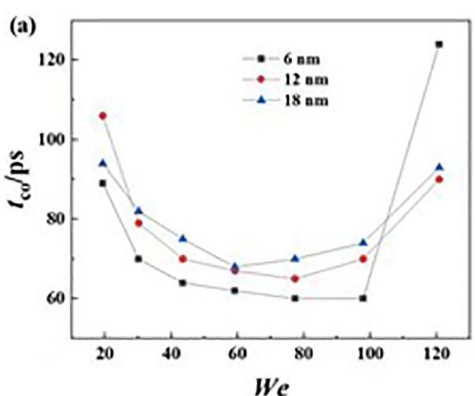

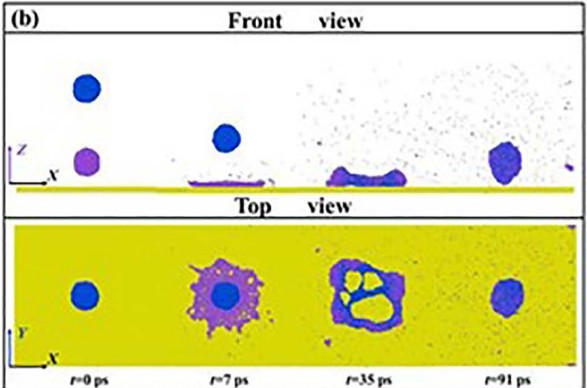

**Fig 10. (a) Variation of $t_c$ on superhydrophobic solid surface under different given conditions. (b)** Evolution of impacting binary droplets at *We*=118.2.

deformation. The $t_c$ gradually reduces to an invariant value; in such a situation, the further increasing $We$ can not further affect $t_c$ because of the impacting droplet being regarded as a rigid film. Intriguingly, the high value of $We$ can increase $t_c$, see $We > 100$ in **Fig 10a**. The corresponding snapshots show that the bouncing pattern changes to the dynamics with a closed-up hole, as shown in **Fig 10b**. The progress of retraction of the hole can significantly increase $t_c$, and thus, the increasing $We$ involves a larger hole and a longer $t_c$. In addition, the increasing $D_{sep}$ shows a tendency to increase $t_c$, owing to the hysteresis of the secondary spreading.

## 4. Conclusions

In this study, the impingement of deposited water nanodroplets with incoming ones with different height differences, $D_{sep}$, via MD simulations. The main conclusions are as follows.

The representative dynamic evolution of impacting binary nanodroplets has been drawn through observation of snapshots extracted from MD simulations. The targeted system first starts to touch the surface and experiences the primary spreading. The primary spreading follows the same step as that for the normal impingement. Subsequently, the upper droplet starts to approach the lower one, but at a different time point. The low-velocity impingement makes coalescence take place at an incomplete retracting state at low $D_{sep}$ value, or at a stable deposited droplet when $D_{sep}$ increases. The high-velocity impingement develops the collision to be advanced to an early retracting stage. For targeted systems, variation of $D_{sep}$ can not change the maximum spreading factor over primary spreading. Therefore, the law of $\beta_{pri,\,max} \sim We^{1/2} Re^{1/5}$ can precisely predict the progress of the primary spreading with a prefactor of 0.06. The $\beta_{sec,\,max}$ can be divided into two parts. Part one is very similar to the progress of the primary spreading with the same prefactor. The increasing $We$ make targeted systems enter the part two. The prefactor is found to be significantly increased from 0.06 to 0.09. This is attributable to the fact that the collision and incomplete retracting of the below droplet accelerate the secondary spreading together with $\beta_{sec,\,max}$. The variation of contact time of targeted systems under different conditions is investigated. The contact time initially reduces as $We$ increases and then plateaus to a constant value, which is induced by the changeable pattern from transmutable droplets to rigid ones. Further increasing $We$ produces different scenes, the increasing $We$ leads to a great increase in the contact time. Through observing snapshots, we find that the collision can induce dynamic instability at the central part. The dynamic instability generates a hole, and the contact time is increased because the closed-up hole needs a much longer time. Therefore, the increasing $We$ in such situations becomes an adverse factor for bouncing of targeted systems.

## Supporting information

**S1 File. Complete data file.**
(ZIP)

## Author contributions

**Conceptualization:** Xudong Ma.

**Data curation:** Liwei Sun.

**Formal analysis:** Liwei Sun, Yi Guo, Tianshi Yang.

**Funding acquisition:** Tianshi Yang, Xudong Ma.

**Investigation:** Yi Guo, Xudong Ma.

**Methodology:** Liwei Sun.

**Project administration:** Yi Guo, Xudong Ma.

**Resources:** Tianshi Yang.

**Software:** Liwei Sun, Tianshi Yang.

**Supervision:** Liwei Sun.

**Validation:** Yi Guo, Xudong Ma.

**Visualization:** Tianshi Yang, Xudong Ma.

**Writing – original draft:** Liwei Sun.

**Writing – review & editing:** Liwei Sun.

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
