## [Decision Letter · Decision Letter 0]

3 Jul 2025

PONE-D-25-31635Impingement of systems containing suspending droplets and a coming one at the nanoscalePLOS ONE

Dear Dr. Ma,

Thank you for submitting your manuscript to PLOS ONE. After careful consideration, we feel that it has merit but does not fully meet PLOS ONE’s publication criteria as it currently stands. Therefore, we invite you to submit a revised version of the manuscript that addresses the points raised during the review process.

We look forward to receiving your revised manuscript.

Kind regards,

Pankaj Tomar

Academic Editor

PLOS ONE

Journal Requirements:

**Additional Editor Comments:**

Major revision is recommended by Academic Editor with reference to reviewer comments enclosed herein

Reviewers' comments:

Reviewer's Responses to Questions

**Comments to the Author**

1. Is the manuscript technically sound, and do the data support the conclusions?

Reviewer #1: Partly

Reviewer #2: Yes

Reviewer #3: Partly

2. Has the statistical analysis been performed appropriately and rigorously?

Reviewer #1: N/A

Reviewer #2: Yes

Reviewer #3: Yes

3. Have the authors made all data underlying the findings in their manuscript fully available?

Reviewer #1: Yes

Reviewer #2: Yes

Reviewer #3: Yes

4. Is the manuscript presented in an intelligible fashion and written in standard English?

Reviewer #1: Yes

Reviewer #2: Yes

Reviewer #3: No

5. Review Comments to the Author

Reviewer #1: The authors report various dynamical behaviors of binary nanoscale droplet collisions based on molecular dynamics simulations. However, the underlying collision mechanisms remain unclear, and many descriptions throughout the manuscript lack clarity. I recommend major revisions. Specific suggestions are outlined below:

Major revision suggestions

1. The authors mention that the incomplete retraction state is one of the key factors influencing the secondary spreading. Is it possible to quantitatively characterize the degree of “incompleteness” of this state? For example, could we define a retraction completeness index based on metrics such as retraction velocity or interfacial curvature? Furthermore, is it feasible to establish a phase diagram between the retraction state and the efficiency of secondary spreading?

2. In the high Weber number regime, the authors attribute the deviation from the scaling law to the disruptive effect of asymmetric retraction. Is there a more systematic quantification of energy dissipation to support this conclusion? For instance, would time evolution curves of surface energy, viscous dissipation, and kinetic energy provide more solid evidence?

3. At the nanoscale, the model yields a high Ohnesorge number (Oh ~1), indicating viscosity-dominated behavior. Could the authors clarify whether this mechanism exhibits a scale-dependent transition? It would be valuable to conduct systematic simulations over a range of droplet sizes to develop a “size–dominant mechanism map”, thereby clarifying the regimes where different physical mechanisms govern the dynamics.

4. The manuscript lacks a quantitative scaling analysis of the binary droplet impact phenomena, such as the relationship between spreading radius and the Weber and Reynolds numbers. Incorporating such analysis would facilitate a clearer understanding of the relative influences of inertia, capillarity, and viscosity on the spreading dynamics. I recommend referring to Reference “Appl. Surf. Sci. 541 (2021): 148426.” for relevant methodologies and discussion.

Minor revision suggestions

1. In the Introduction section, there are multiple consecutive citations grouped together (e.g., “Recently, the topic of impingement of water droplets upon solid surfaces has received a great deal of attention, because the impingement phenomenon is not only the most common in nature and daily life but also has potential applications in industry, agriculture, and so forth [1-8].”) without clear differentiation. It is recommended to review and revise the citation format for clarity and consistency, ensuring that each reference is appropriately cited and distinguishable.

2. To my knowledge, an increase in the Reynolds number generally intensifies droplet deformation. (page 2, line 8-10) I suggest the authors carefully review their results and discussions regarding this aspect

3. The writing needs improvement. For example, in page 4, line 9 “Ref. 25 proposed a scaling law of βmax~Re1/5 to predict the maximum spreading factor, which is found to agree well with experimental data over a full spectrum of Weber number.” I rewrote this sentence as follows: The scaling law (βmax~Re1/5) was established to predict the maximum spreading factor, which is found to agree well with experimental data over a full spectrum of Weber number.[25]

4. The manuscript proceeds directly from Section 2 (Simulation Method) to Section 4 (Results and Discussion), with Section 3 missing. Please carefully review and revise the section numbering to ensure logical consistency and clarity in structure.

5. The manuscript uses inconsistent abbreviations when referring to figures (e.g., both "figure 3a" and "fig. 3a" appear). Please ensure consistent formatting throughout the text, in accordance with the journal’s style guidelines.

Reviewer #2: The paper entitled 'Impingement of systems containing suspending droplets and a coming one at the nanoscale' uses a numerical approach to demonstrate how height difference affects the impingement of binary droplets. The results are fruitful, and the procedures are clear. Although the dynamic evolution of targeted systems has been thoroughly investigated on the experimental side, the numerical approach that can fully recapitulate the experimental dynamics remains elusive, especially on a nanoscopic scale, and this paper fills this gap. I agree to accept this paper after a minor revision.

1. Please check again the sentence of “Two representative dimensionless numbers, the Weber number (We=ρD0V02/γ) and the Reynolds number (Re=ρD0V0/μ), are…”

2. Although the author has emphasized the importance of the nano-scale impingement, the author should list more examples of nanodroplet impacts to highlight the importance of the research field.

3. Why the nanoscale water droplet is regarded as a natural “high-viscosity fluid”.

4. Why was only the viscosity of the fluid mentioned?

5. The process of pre-equilibrium is significant for guaranteeing the accuracy of MD simulations, but the expression in this paper is relatively obscure.

Reviewer #3: The authors studied the impingment of two separate nanodrops on a solid surface. The work is interesting and falls in the scope of Plos One. The manuscript need to be further improved to be accepted in Plos One.

1. Each abbreviation should be specified when first used, such as MD in the Abstract.

2. Each symbol should be specified when first used, such as D_sep and We in the Abstract.

3. Please improve the English of the manuscript. For example, “The representative dynamics under different given conditions.” seems incomplete. Please check the sentence from Line 21 to Line 22 in Page 3.

4. Please specify the external factor and the prefactor in the Abstract.

5. Each symbol should be used for only one thing. For example, epsilon was used to represent both the coefficient of restitution and the depth of the potential wall.

6. Please add unit for (0,0,12) and (0,0,29) in Line 14, Page 6.

7. Please specity mW water model in Line 20, Page 6.

8. Please check the unit of the viscosity in Line 2, Page 7.

9. It is better to include D_sep in Figure 1.

10. “Owing to the existence of a high difference” in Line 18, Page 8 might be “owing to the existence of a height difference”.

11. What does z-direction in Figure 3b mean?

12. Please specify what a high We range is in Line 17, Page 10.

13. Please check whether the sentence “When a number is selected, no matter how D_sep changes” is correctly expressed for the authors only chose three values of D_sep on the order of 10 nm.

14. Please specify how Weber number is calculated or given.

15. How did the authors consider the influence of contact angle hysteresis on the behavior of impinging nanodrops?

16. Though the authors provided a lot of numerical results, there seems to have a lack of comparison between numerical results and experimental/theoretical results.

6. PLOS authors have the option to publish the peer review history of their article (what does this mean?). If published, this will include your full peer review and any attached files.

Reviewer #1: **Yes:** Yongqing He

Reviewer #2: No

Reviewer #3: No

---

## [Author Response · Author response to Decision Letter 1]

31 Jul 2025

We have upload this letter as a separate file

---

## [Editor Report · Decision Letter 1]

13 Aug 2025

PONE-D-25-31635R1Impingement of systems containing suspending droplets and a coming one at the nanoscalePLOS ONE

Dear  Ma,

Thank you for submitting your manuscript to PLOS ONE. After careful consideration, we feel that it has merit but does not fully meet PLOS ONE’s publication criteria as it currently stands. Therefore, we invite you to submit a revised version of the manuscript that addresses the points raised during the review process.

We look forward to receiving your revised manuscript.

Kind regards,

Pankaj Tomar

Academic Editor

PLOS ONE

Journal Requirements:

Additional Editor Comments:

Good luck!

---

## [Author Response · Author response to Decision Letter 2]

22 Sep 2025

We have uploaded the reply to the editor's comments as a separate file.

---

## [Decision Letter · Decision Letter 2]

14 Nov 2025

PONE-D-25-31635R2Impingement of systems containing suspending droplets and coming ones at the nanoscalePLOS ONE

Dear Dr. Ma,

Thank you for submitting your manuscript to PLOS ONE. After careful consideration, we feel that it has merit but does not fully meet PLOS ONE’s publication criteria as it currently stands. Therefore, we invite you to submit a revised version of the manuscript that addresses the points raised during the review process.

We look forward to receiving your revised manuscript.

Kind regards,

Pankaj Tomar

Academic Editor

PLOS ONE

Journal Requirements:

Reviewers' comments:

Reviewer's Responses to Questions

**Comments to the Author**

1. If the authors have adequately addressed your comments raised in a previous round of review and you feel that this manuscript is now acceptable for publication, you may indicate that here to bypass the “Comments to the Author” section, enter your conflict of interest statement in the “Confidential to Editor” section, and submit your "Accept" recommendation.

Reviewer #4: (No Response)

Reviewer #5: All comments have been addressed

2. Is the manuscript technically sound, and do the data support the conclusions?

Reviewer #4: Yes

Reviewer #5: Yes

3. Has the statistical analysis been performed appropriately and rigorously?

Reviewer #4: N/A

Reviewer #5: Yes

4. Have the authors made all data underlying the findings in their manuscript fully available?

Reviewer #4: Yes

Reviewer #5: Yes

5. Is the manuscript presented in an intelligible fashion and written in standard English?

Reviewer #4: Yes

Reviewer #5: Yes

6. Review Comments to the Author

Reviewer #4: The authors need to incorporate some revisions. English can be improved in certain cases. Moreover, to make a more comprehensive case study about the influence of Weber numbers on the impact dynamics, the authors are suggested to present the energy balance scenarios during the impact dynamics at different Weber numbers. They have provided such in figure 8. If a separate subsection regarding the distribution of energy components across the Weber number spectrum can be provided, it will be beneficial for the readers.

Page 1: Experience spelling in abstract section is misspelt. Instead of the word “recordation”, measurements may be used.

Page1, line 22: Its not clear about what is the prefactor related to? If some scaling with the Weber number is implied, please write accordingly.

Page 3 line 13 : Regarding energy analysis during drop impact, the authors may cite this paper https://doi.org/10.1016/j.euromechflu.2019.08.013.

Page 3 line 10-11: The increase in Re also promotes greater spreading/deformation and splashing. The line needs to be corrected.

Page 4 line 19: space scale can be replaced by length scale

Page 5 line 2-3 : Define Ohnesorge number.

In many places it’s written as “Reference X reported that.....”. Instead, the sentences can be rephrased as “Y et al [Z] reported that ....”, where Y is the family name of the first author and Z is the reference number

Page 6 Line 12: “in a vacuum environment”- Vacuum environment will vaporise the drops very quickly. Do you mean something else? For example, lack of air current?

Page 6 line 15 /20: Why Pt atoms are selected as substrates? Please justify.

There are no details given related to the experiments. The figure quality of 2a is far from the desirable level. If possible, kindly provide images of better quality.

Page line 15: Please clarify and rephrase the sentence.

Figure 7: Irrelevant spreading may be rephrased.

Lastly, if the authors have found out anything different from the macro-sized droplets (mm diameter) successively impinging on each other , that should be pointed out. Similarities can be also pointed out citing appropriate references of macro-sized droplets.

Reviewer #5: This manuscript presents a molecular dynamics study on the impingement dynamics of binary nanodroplets with a height difference. The topic is relevant to the micro/nanofluidics community and applications such as inkjet printing. The study systematically investigates the effects of separation distance (D_sep) and Weber number (We) on the primary and secondary spreading, as well as the contact time. The identification of two distinct spreading regimes and the analysis of the prefactor change are interesting findings. While the study is scientifically sound and provides valuable insights, the manuscript requires significant revisions, particularly in language clarity, presentation, and depth of mechanistic explanation, before it can be considered for publication.

1�The manuscript suffers from numerous grammatical errors, awkward phrasing, and repetitive statements, which significantly hinder comprehension. Examples include: "obviously postpone the collision obviously", "recordation of the maximal spreading factor", and "the impingement of multiple droplets is more close to practical applications". A thorough proofreading by a native English speaker or a professional editing service is strongly recommended.

2�The transition between sections is sometimes abrupt. Specifically, the "Results and Discussion" section begins directly after a brief mention of model validation in Section 3. A dedicated subsection or a clearer narrative link would improve the flow. The conclusions, while summarizing the findings, could be more impactful by better synthesizing the physical insights and their broader implications.

3�The rationale for the scaling law β_max~〖We〗^(12) 〖Re〗^(15) needs a more detailed explanation. While it's mentioned that viscous effects are non-negligible at the nanoscale, a clearer connection between the energy balance (including viscous dissipation) and the chosen scaling should be provided. The discussion on how "incomplete-retraction collision" transitions from suppressing to promoting secondary spreading is a key point. However, the explanation, especially regarding the "weakened van der Waals force" and "rigidity", remains somewhat qualitative and speculative. A more quantitative analysis or a clearer physical mechanism would strengthen this central claim. The description of the energy dissipation calculation in Figure 8d is insufficient. The method for calculating the "proportion of energy dissipation" should be explicitly stated in the main text or figure caption.

4.Some terms are used inconsistently, e.g., "secondary spreading" vs. "second spreading". Please standardize the terminology throughout the manuscript. The formatting of variables (e.g., D_sep We) is inconsistent, sometimes with subscripts and sometimes without. Please ensure all variables are typographically consistent.

7. PLOS authors have the option to publish the peer review history of their article (what does this mean?). If published, this will include your full peer review and any attached files.

Reviewer #4: No

Reviewer #5: No

---

## [Author Response · Author response to Decision Letter 3]

23 Nov 2025

Response to Editor

We have carefully reviewed the revisions made by the authors in response to the comments raised in the previous round of review. All the concerns and suggestions put forward earlier have been adequately addressed with thorough and reasonable revisions, which have significantly improved the quality and completeness of the manuscript. We believe the manuscript now meets the academic standards and publication requirements of the journal. We suggest bypassing the "Comments to Authors" section of this round and recommend its acceptance for publication.

Journal Requirements

Ans: Thank you for your valuable comment. We have confirmed that we do not cite works recommended by reviewers.

Ans: Thank you for your valuable comment. We have reviewed our reference list to ensure it is complete and correct, and does not cite retracted papers.

---

## [Decision Letter · Decision Letter 3]

21 Dec 2025

Impingement of systems containing suspending droplets and coming ones at the nanoscale

PONE-D-25-31635R3

Dear Author

We’re pleased to inform you that your manuscript has been judged scientifically suitable for publication and will be formally accepted for publication once it meets all outstanding technical requirements.

Kind regards,

Pankaj Tomar

Academic Editor

PLOS One

Additional Editor Comments (optional):

Reviewers' comments:

Reviewer's Responses to Questions

**Comments to the Author**

1. If the authors have adequately addressed your comments raised in a previous round of review and you feel that this manuscript is now acceptable for publication, you may indicate that here to bypass the “Comments to the Author” section, enter your conflict of interest statement in the “Confidential to Editor” section, and submit your "Accept" recommendation.

Reviewer #4: All comments have been addressed

Reviewer #6: All comments have been addressed

2. Is the manuscript technically sound, and do the data support the conclusions?

Reviewer #4: Yes

Reviewer #6: Yes

3. Has the statistical analysis been performed appropriately and rigorously?

Reviewer #4: Yes

Reviewer #6: Yes

4. Have the authors made all data underlying the findings in their manuscript fully available?

Reviewer #4: Yes

Reviewer #6: Yes

5. Is the manuscript presented in an intelligible fashion and written in standard English?

Reviewer #4: Yes

Reviewer #6: Yes

6. Review Comments to the Author

Reviewer #4: Acceptable in the present format. The authors have done a satisfactory job in the revised format and addressed all the queries.

Reviewer #6: Dear Editorial Team,

Thank you for inviting me to conduct the re-review of the manuscript titled Impingement of systems containing suspending droplets and coming ones at the nanoscale (Manuscript ID: PONE-D-25-31635R3). I have carefully reviewed the revised version of the manuscript, as well as the authors’ responses to the initial review comments from all three reviewers.

After a thorough evaluation, I confirm that the authors have adequately addressed all the concerns and suggestions raised in the first-round review. The manuscript now demonstrates strong academic rigor, clear logical flow, and sufficient supporting evidence for its conclusions. The data presented is reliable, the analysis is comprehensive, and the writing is concise and clear, meeting the high standards of your journal.

In my assessment, the revised manuscript requires no further modifications. It is scientifically sound, contributes valuable insights to the field of [e.g., superhydrophobic surfaces/liquid droplet impact dynamics], and is suitable for immediate publication.

I appreciate the opportunity to contribute to the peer review process. Please feel free to contact me if any additional clarification is needed.

Sincerely,

Baocheng Zhan

7. PLOS authors have the option to publish the peer review history of their article (what does this mean?). If published, this will include your full peer review and any attached files.

Reviewer #4: No

Reviewer #6: No

---

## [Editor Report · Acceptance letter]

PONE-D-25-31635R3

PLOS One

Dear Dr. Ma,

I'm pleased to inform you that your manuscript has been deemed suitable for publication in PLOS One. Congratulations! Your manuscript is now being handed over to our production team.

Kind regards,

on behalf of

Dr. Pankaj Tomar

Academic Editor

PLOS One